# Dex-NeRF: Using a Neural Radiance Field to Grasp Transparent Objects

**Jeffrey Ichnowski** *
The AUTOLAB
University of California Berkeley, USA
jeffi@berkeley.edu

**Yahav Avigal** *
The AUTOLAB
University of California, Berkeley, USA
yahav_avigal@berkeley.edu

**Justin Kerr**
The AUTOLAB
University of California, Berkeley, USA
justin_kerr@berkeley.edu

**Ken Goldberg**
The AUTOLAB
University of California, Berkeley, USA
goldberg@berkeley.edu

**Abstract:** The ability to grasp and manipulate transparent objects is a major challenge for robots. Existing depth cameras have difficulty detecting, localizing, and inferring the geometry of such objects. We propose using neural radiance fields (NeRF) to detect, localize, and infer the geometry of transparent objects with sufficient accuracy to find and grasp them securely. We leverage NeRF's view-independent learned density, place lights to increase specular reflections, and perform a transparency-aware depth-rendering that we feed into the Dex-Net grasp planner. We show how additional lights create specular reflections that improve the quality of the depth map, and test a setup for a robot workcell equipped with an array of cameras to perform transparent object manipulation. We also create synthetic and real datasets of transparent objects in real-world settings, including singulated objects, cluttered tables, and the top rack of a dishwasher. In each setting we show that NeRF and Dex-Net are able to reliably compute robust grasps on transparent objects, achieving 90 % and 100 % grasp-success rates in physical experiments on an ABB YuMi, on objects where baseline methods fail. See https://sites.google.com/view/dex-nerf for code, video, and datasets.

## 1 Introduction

Transparent objects are common in homes, restaurants, retail packaging, labs, gift shops, hospitals, and industrial warehouses. Effectively automating robotic manipulation of transparent objects could have a broad impact, from helping in everyday tasks and performing tasks in hazardous environments. Existing depth cameras assume that surfaces of observed objects reflect light uniformly in all directions, but this assumption does not hold for transparent objects as their appearance varies significantly under different view directions and illumination conditions due to reflection and refraction properties of transparent materials. In this paper, we propose and demonstrate *Dex-NeRF*, a new method to sense the geometry of transparent objects and allow for robots to interact with them.

Dex-NeRF uses a Neural Radiance Fields (NeRF) as part of a pipeline (Fig. 1, right) to compute and execute robot grasps on transparent objects. While NeRF was originally proposed as an alternative for explicit volumetric representations and shown to render novel views of complex scenes realistically [1], it can also reconstruct the scene geometry. In particular, due to the view-dependent nature of the NeRF model, it can learn to represent the geometry associated with transparency accurately. The only input requirement to train a NeRF model is a set of images taken from a camera with known intrinsics (e.g., focal length, distortion) and extrinsics (position and orientation in the world). While the intrinsics can be determined from calibration techniques or from the camera itself, determining the extrinsics is often a challenge [2, 3]. However, robots operating in a fixed workcell or with a camera mounted on the manipulator arm, can readily determine camera intrinsics. This makes NeRF a particularly good match for robot manipulators.

---

*Equal contribution.

5th Conference on Robot Learning (CoRL 2021), London, UK.

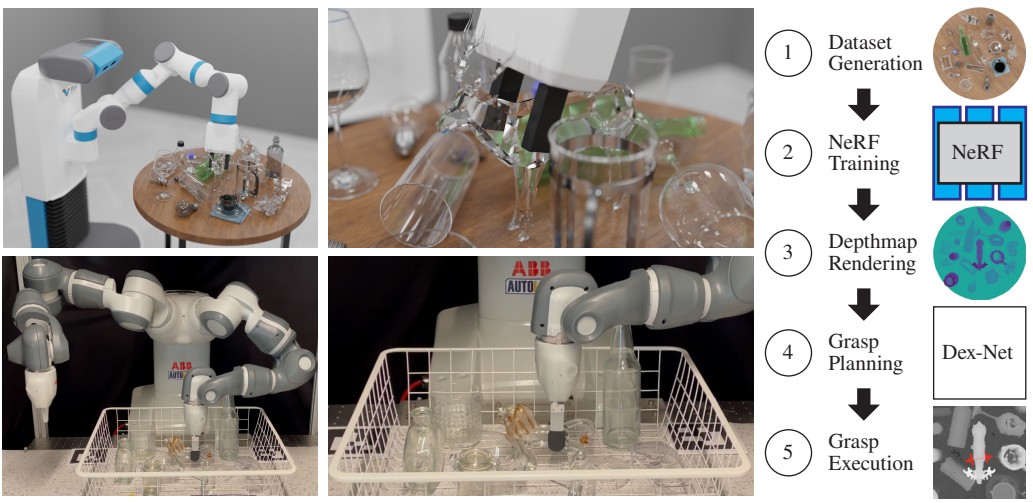

Figure 1: **Using NeRF to grasp transparent objects** Given a scene with transparent objects (left column), we the pipeline on the right to compute grasps (middle column). The top row shows Dex-NeRF working in a simulated scene while the bottom row shows it working in a physical scene.

In experiments, we show qualitatively and quantitatively that Dex-NeRF can compute high accuracy depth images from photo-realistic synthetic and real scenes, and achieve 90 % or better grasp-success rates on real objects.

The contributions of this paper are: (1) integration of NeRF with robot grasp planning, (2) a transparency-aware depth rendering method for NeRF, (3) experiments on synthetic and real images showing NeRF with Dex-Net generates high-quality grasps, (4) synthetic and real image datasets with transparent objects for training NeRF models.

## 2   Related Work

**Detecting Transparent Objects**   For robots to interact with transparent objects, they must first be able to detect them.  The most recent approaches detecting and recognizing transparent objects are data-driven. Lai et al. [4] and Khaing et al. [5] propose using a Convolutional Neural Network (CNN) to detect transparent objects in RGB images. Recently, Xie et al. [6] developed a transformer-based pipeline [7] for transparent object segmentation. Other methods rely on deep-learning models to predict the object pose. Phillips et al. [8] trained a random forest to detect the contours of transparent objects for pose estimation and shape recovery. Xu et al. [9] proposed a two-stage method for estimating the 6-degrees-of-freedom (DOF) pose of a transparent object from a single RGBD image by replacing the noisy depth values with estimated values and training a DenseFusion-like network structure [10]. Sajjan et al. [11] extend this and incorporate a neural network trained for 3D pose estimation of transparent objects in a robotic picking pipeline. Zhou et al. [12, 13] train a grasp planner directly on raw images from a light-field camera. Zhu et al. [14] used an implicit function to complete missing depth given noisy RGBD observation of transparent objects. However, these data-driven methods rely on large annotated datasets that are hard to curate, whereas Dex-NeRF does not require any prior dataset.

**Neural Radiance Fields**   Recently, implicit neural representations have led to significant progress in 3D object shape representation [15, 16, 17] and encoding the geometry and appearance of 3D scenes [18, 1]. Mildenhall et al. [1] presented Neural Radiance Fields (NeRF), a neural network whose input is a 3D coordinate with an associated view direction, and output is the volume density and view-dependent emitted radiance.  Due to its view-dependent prediction, NeRF can represent non-Lambertian effects such as specularities and reflections, and therefore capture the geometry of transparent objects.  However, NeRF is slow to train and has low data efficiency.  Yu et al. [19] proposed *Plenoctrees*, mapping coordinates to spherical harmonic coefficients, shifting the view-dependency from the input to the output.  In addition, Plenoctrees pre-samples the model into a sparse octree structure, achieving a significant speedup in training over NeRF. Deng et al. [20] proposed JaxNeRF, an efficient JAX implementation of NeRF reduces the training time of a NeRF model from over a day to several hours. Deng et al. [21] add depth supervision to train NeRF 2 to

$6\times$ faster given fewer training views. Adamkiewicz et al. [22] proposed an algorithm that uses a NeRF model for robot navigation. In this work, we propose to use NeRF to recover the geometry of transparent objects for the purpose of robotic manipulation.

**Robotic Grasping** Traditional robot grasping methods analyze the object shape to identify successful grasp poses [23, 24, 25]. Data-driven approaches learn a prior using labeled data [26, 27] or through self-supervision over many trials in a simulated or physical environment [28, 29] and generalize to grasping novel objects with unknown geometry. These approaches rely on RGB and depth sensors to generate an accurate observation of the target object. Additionally, different methods use different inputs, such as depth maps [30, 31, 32], point clouds [33, 34, 35, 9], octrees [36], or a truncated signed distance function (TSDF) [37, 38]. In contrast, in this paper we propose a method to render a high-quality depth map from a NeRF model to then pass to Dex-Net [30] to compute a grasp. While standard depth cameras have gaps in their depth information that needs to be processed out with hole-filling techniques, the depth map rendering from NeRF is directly usable. It is possible that other grasp-planning techniques may be able to plan grasps from NeRF models.

## 3 Problem Statement

We assume an environment with an array of cameras at known fixed locations or that the robot can manipulate a camera (e.g., wrist-mounted) to capture multiple images of the scene. Given the environment with rigid transparent objects, Dex-NeRF computes a frame for a robot gripper that will result in a stable grasp of a transparent object.

## 4 Method

This section provides a brief background on NeRF, then describes recovering geometry of transparent objects, integrating with grasp analysis, and improving performance with additional lights.

### 4.1 Preliminary: Training NeRF

NeRF [1] learns a neural scene representation that maps a 5D coordinate containing a spatial location $(x, y, z)$ and viewing direction $(\theta, \phi)$ to the volume density $\sigma$ and RGB color $\mathbf{c}$. Training NeRF's multilayer perceptron (MLP) requires multiple RGB images of a static scene with their corresponding camera poses and intrinsic parameters. The expected color $C(\mathbf{r})$ of the camera ray $\mathbf{r} = \mathbf{o} + t\mathbf{d}$ between near and far scene bounds $t_n$ and $t_f$ is:

$$C(\mathbf{r}) = \int_{t_n}^{t_f} T(t)\sigma(\mathbf{r}(t))\mathbf{c}(\mathbf{r}(t), \mathbf{d})dt, \tag{1}$$

where $T(t) = \exp\left(-\int_{t_n}^{t} \sigma(\mathbf{r}(s))ds\right)$ is the probability that the camera ray travels from near bound $t_n$ to point $t$ without hitting any surface. NeRF approximates the expected color $\hat{C}(\mathbf{r})$ as:

$$\hat{C}(\mathbf{r}) = \sum_{i=1}^{N} T_i(1 - \exp(-\sigma_i\delta_i))\mathbf{c}_i, \tag{2}$$

where $T_i = \exp\left(-\sum_{j=1}^{i-1} \sigma_j\delta_j\right)$ and $\delta_i = t_{i+1} - t_i$ is the distance between consecutive samples on the ray $\mathbf{r}$. The training process minimizes the error between rendered and ground-truth colors.

### 4.2 Recovering Geometry of Transparent Objects

We observe that NeRF does not directly support transparent object effects—it casts a single ray per source image pixel without reflection, splitting, or bouncing. NeRF recovers non-Lambertian effects such as reflections from a specular surface by regressing on view direction and supervising with view-dependent emitted radiance. However, while RGB color $\mathbf{c}$ is view-dependent, the volume density $\sigma$ is not—meaning NeRF has to learn a non-zero $\sigma$ to represent any color at that spatial location. The usual result is that the transparent object shows up as a "ghostly" or "blurry" version of the original object in rendered RGB images.

When training, a NeRF model learns a density $\sigma$ of each spatial location. This density corresponds to the transparency of the point, and serves to help learn how much a spatial location contributes to

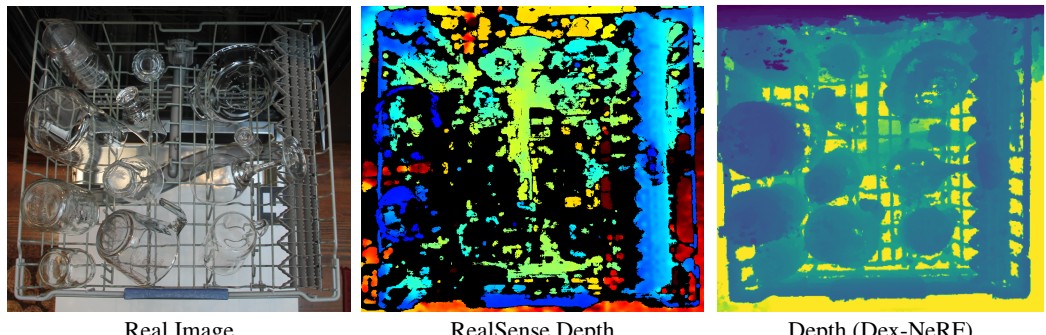

| Real Image | RealSense Depth | Depth (Dex-NeRF) |
|---|---|---|

Figure 2: **Comparison to RealSense Depth Camera**. We compare the results of the proposed pipeline in a real-world setting against the depth map produced by an Intel RealSense camera. In the left image is the real-world scene, the middle shows the depth image from the RealSense, and the right shows the result of our pipeline. The color scheme in the RealSense image is provided by the RealSense SDK, while the color scheme in the right column is from MatPlotLib. We observe that the RealSense depth camera is unable to recover depth from a large portion of the scene, shown in black. On the other hand, the proposed pipeline, while having a few holes, can recover depth for most of the scene.

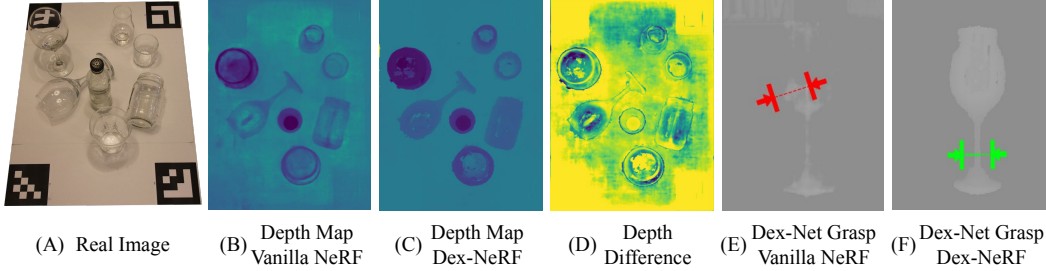

(A) Real Image    (B) Depth Map Vanilla NeRF    (C) Depth Map Dex-NeRF    (D) Depth Difference    (E) Dex-Net Grasp Vanilla NeRF    (F) Dex-Net Grasp Dex-NeRF

Figure 3: **Using NeRF to render depth for grasping transparent objects**. Dex-NeRF uses a transparency-aware depth rendering to render depth maps that can be used for grasp planning. In contrast, Vanilla-NeRF's depth maps are filled with holes and result in poor grasp predictions.

the color of a ray cast through it. Although NeRF converts each $\sigma_i$ to an occupancy probability $\alpha_i = 1 - \exp(-\sigma_i \delta_i)$, where $\delta_i$ is the distance between integration times along the ray, thus implicitly giving $\alpha_i$ an upper bound of 1, it does not place a bound on the raw $\sigma$ value. Dex-NeRF uses the raw value of $\sigma$ to determine if a point in space is occupied.

### 4.3 Rendering Depth for Grasp Analysis

To compute a grasp from a trained NeRF model, we propose to render a depth image and have Dex-Net use it to plan the grasp. To generate a depth image, we consider two candidate reconstructions of depth. First, we use the same depth rendering that NeRF uses. This *Vanilla NeRF* reconstruction first converts $\sigma_i$ to an occupancy probability $\alpha_i$. It then applies the transformation $w_i = \alpha_i \prod_{j=1}^{i-1} (1 - \alpha_j)$. To render depth at pixel coordinate $[u, v]$, it computes the sum of sample distances from the camera weighted by the termination probability $D[u, v] = \sum_{i=1}^{N} w_i \delta_i$. When applied on transparent objects, however, this results in noisy depth maps, as shown in Fig. 3.

Instead, we consider a second, transparency-aware method that searches for the first sample along the ray for which $\sigma_i > m$, where $m$ is a fixed threshold. The depth is then set to the distance of that sample $\delta_i$. We explore different values for $m$, and observe that low values result in a noisy depth map while high values create holes in the depth map. In our experiments we set $m = 15$ (see Fig. 8).

### 4.4 Improving Reconstruction with Light Placement

For NeRF to learn the geometry of a transparent object, it must be able to "see" it from multiple camera views. If the transparent object is not visible from any views, then it will have no effect on the loss function used in training, and thus not be learned. We thus look for a way to improve visibility of transparent objects to NeRF.

One property that transparent objects share (e.g., glass, clear plastic) is that they are glossy and thus produce specular reflections when the camera view direction is opposite to the surface normal of the incident direction of light. To the NeRF model, a specular reflection viewed from multiple angles will appear as a bright point on a solid surface—e.g., $\mathbf{c} = [1, 1, 1]^T$ and $\sigma > 0$, while from other angles it will appear as $\sigma \leq 0$. As $\sigma$ is view-independent, NeRF learns a $\sigma$ between fully opaque and fully transparent for such points.

By placing additional lights in the scene, we create more angles from which cameras will see specular reflections from transparent objects—this results in NeRF learning a model that fills holes in the scene. While the number and placement of lights for optimal training is dependent on both the expected object distribution and camera placement, in experiments (Sec. 5.5) we show that increasing from 1 light to a 5x5 array of lights improves the quality of the learned geometry.

## 5 Experiments

We experiment in both simulation and on a physical ABB YuMi robot. We generate multiple datasets, where each dataset consists of images and associated camera transforms of one static scene containing one or more transparent objects. We train NeRF models using a modified JaxNeRF [20] implementation on 4 Nvidia V100 GPUs. We use an existing pre-trained Dex-Net model for grasp planning without modification or fine-tuning. We can do this since NeRF models can be rendered to depth maps from arbitrary camera intrinsics and extrinsics, thus we match our NeRF rendering to the Dex-Net model instead of training a new one.

### 5.1 Datasets

As existing NeRF datasets do not include transparent objects, and existing transparent-object-grasping datasets do not include multiple camera angles, we generate new datasets using 3 different methods: synthetic, Cannon EOS 60D camera with a Tamron Di II lens with a locked focal length, and an Intel RealSense.

For synthetic datasets, we use Blender 2.92's physically-based Cycles renderer with path tracing set to 10240 samples per pixel, and max light path bounces set to 1024. We chose theses settings by increasing them until renderings were indistinguishable from the previous setting—finding that lower settings lead to dark regions and smaller specular reflections. For glass materials, we set the index of refraction to 1.45 to match physical glass. We include 8 synthetic datasets of transparent objects: 2 scenes with clutter: light array and single light; 4 singulated objects from Dex-Net: Pipe Connector, Pawn, Turbine Housing, Mount; and 2 household objects: Wineglass upright and Wineglass on side. As these computationally demanding to render due to the high quality settings, we distribute these as part of the contribution.

For the Cannon EOS and RealSense real-world datasets, we place ArUco markers in the scene to aid in camera pose recovery and take photos around the objects using a fixed ISO, f-stop, and focal length. We use bundle adjustment from COLMAP [2, 3] to refine the camera poses and intrinsics to high accuracy. We include 8 physical datasets of transparent objects with a variety of camera poses: table with clutter, Dishwasher, Tape Dispenser, Wineglass on side, Flask, Safety Glasses, Bottle upright, Lion Figurine in clutter. The main difficulty in generating these datasets is calibration and computing high-precision camera poses.

The datasets (at https://sites.google.com/view/dex-nerf) differ from prior work in their focus on scenes with transparent objects in a graspable setting, with over 70 camera poses each.

### 5.2 Synthetic Grasping Experiments

We test the ability of Dex-NeRF to generate grasps on the synthetic singulated transparent Dex-Net object datasets. For each dataset, we evaluate the grasp in simulation using a wrench resistance metric measuring the ability of the grasp to resist gravity [39]. Fig. 4 shows images of the synthetic objects, Dex-NeRF-generated depth map, and an example sampled grasp for each. To measure the effect of training time on grasp-success rate, we simulate and record grasps over the course of training. In Fig. 5, we observe that grasp-success rate improves with training time, but plateaus between 80 % and 98 % success rate at around 50k to 60k iterations. This suggests that there may be a practical fixed iteration limit to obtain high grasp success rates.

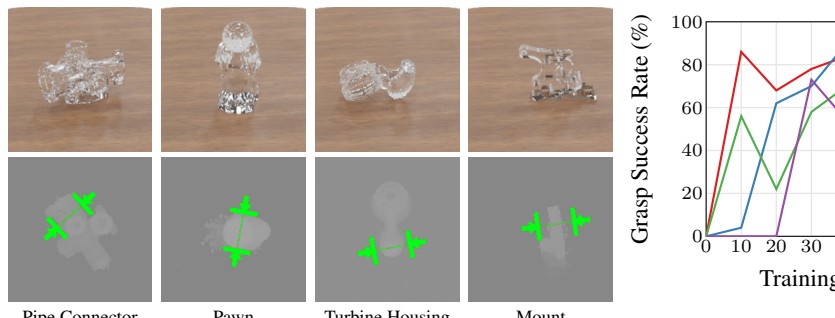

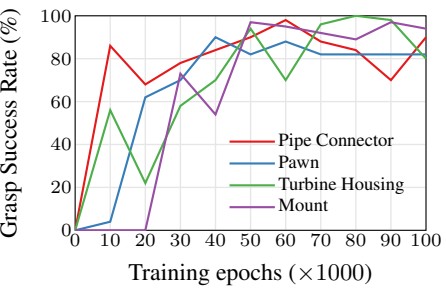

Figure 4: **Synthetic singulated objects** used in simulation experiments. **Top row:** image of the object in the training data. **Bottom row:** computed depth map and candidate grasp.

Figure 5: **Grasp-success rate vs training epochs.** As opposed to view-synthesis, which requires over 200k epochs, we observe high grasp success rates after 50k to 60k epochs.

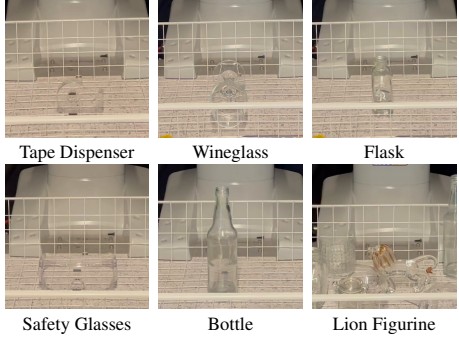

Figure 6: **Physical grasps objects.** In the background is the base of the YuMi robot.

| Object | PhoXi | Vanilla NeRF | Dex-NeRF |
|---|---|---|---|
| Tape Dispenser | 0/10 | 0/10 | **10/10** |
| Wineglass | 0/10 | 0/10 | **9/10** |
| Flask | 0/10 | 1/10 | **9/10** |
| Safety Glasses | 0/10 | 0/10 | **10/10** |
| Bottle | 0/10 | **10/10** | **10/10** |
| Lion Figurine | 0/10 | 3/10 | **10/10** |

Table 1: **Physical grasp success rate.** For each object, we compute a depth map using a PhoXi camera, unmodified Vanilla NeRF, and Dex-NeRF for grasping transparent objects. From the depth map, Dex-Net computes a 10 different grasps, and an ABB YuMi attempts the grasp. Successful grasps lift the object.

We test Dex-NeRF on a scene of a tabletop cluttered with transparent objects. In this experiment, the goal is to grasp a transparent object placed in a stable pose in close proximity to other transparent objects. The challenge is twofold: the depth rendering quality should be sufficient for both grasp planning and collision avoidance. Fig. 1 shows the robot and scene in the upper left, and the overhead image, depth, and computed grasp inline in the pipeline, and the final computed grasp with simulated execution is in the upper middle image. The final grasp contact point was accurate to a 2 mm tolerance, suggesting that Dex-NeRF with sufficient images taken from precisely-known camera locations may be practical in highly cluttered environments.

## 5.3 Physical Grasping Experiments

To test the Dex-NeRF in a physical setup, we place transparent singulated objects in front of an ABB YuMi robot, and have the robot perform the computed grasps. We compare to 2 baselines: (1) *PhoXi*, in which a PhoXi camera provides the depth map; and (2) *Vanilla NeRF*, in which we use the original depth rendering from NeRF. The PhoXi camera is normally able to generate high-precision depth maps for non-transparent objects. All methods use the same pre-trained Dex-Net model, and both Vanilla NeRF and Dex-NeRF use the same NeRF model—the only difference is the depth rendering. We test with 6 objects (Fig. 6), and compute and execute 10 different grasps for each and record the success rate. A grasp is successful if the robot lifts the object. In Table 1, we see that Dex-NeRF gets 90 % and 100 % success rates for all objects, while the baselines get few successful grasps. The PhoXi camera is unable to recover any meaningful geometry which causes Dex-Net predictions to fail. The Vanilla NeRF depth maps often have unpredictable protrusions that result in Dex-Net generating unreliable grasps.

## 5.4 Comparison to RealSense Depth

We qualitatively compare the rendered depth map of the proposed pipeline against a readily-available depth sensor on scenes with transparent objects in real-world settings (Fig. 2). We select the Intel RealSense as it is common to robotics applications, readily available, and high-performance. The

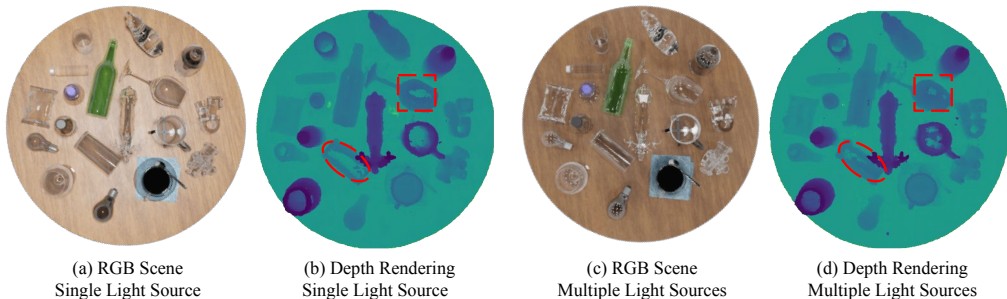

(a) RGB Scene
Single Light Source

(b) Depth Rendering
Single Light Source

(c) RGB Scene
Multiple Light Sources

(d) Depth Rendering
Multiple Light Sources

Figure 7: More lights mean more specular reflections, and result in better NeRF depth estimation of transparent surfaces. In (a) and (b), we show a scene lit by a single overhead high-intensity light. In (c) and (d) we show the same scene lit by an overhead 5x5 array of lights. The combined light wattage is equal in both scenes. Images (a) and (c) are views of the scene, and (b) and (d) are the corresponding depth images obtained from the pipeline. Two glasses on their sides are missing top surfaces (outlined in dashed red) in (b), while the effect is reduced in (d) due to the additional light sources.

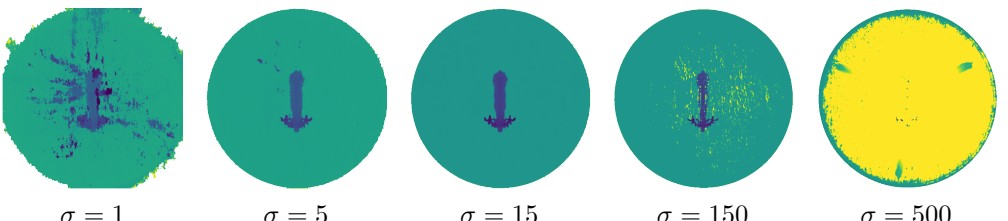

$\sigma = 1$        $\sigma = 5$        $\sigma = 15$        $\sigma = 150$        $\sigma = 500$

Figure 8: **depth rendering using NeRF with different thresholds** Here we show the effect of the threshold value on the depth rendering of an isolated deer figurine. Values too low result in excess noise, while values too high cause parts of the scene to disappear.

RealSense, like most stereo depth cameras, struggles with transparent objects as they are unable to compute a stereo disparity between pixels from different cameras when the pixels are specular reflections or the color of the object behind the transparent object. The RealSense optionally projects a structured light pattern on the scene to aid in computing depth from textureless surfaces; however, in experiments, we observed no qualitative difference with and without the light pattern emitter enabled. We use a Canon EOS for NeRF, and use a RealSense for a depth image. In this experiment, we observe that the RealSense cannot compute the depth of most transparent objects and often produces regions of unknown depth (shown in black) where transparent objects are. On the other hand, the proposed pipeline produces high-quality depth maps with only a few noisy areas.

## 5.5 One vs Many Lights

We experiment with different light setups to test the effect of specular reflections on the ability of NeRF to recover the geometry of transparent objects. We create two scenes (Fig. 7), one with a single bright light source directly above the work surface, and another with an array of 5x5 (25) lights above the work surface. We set the total wattage of the lights in each scene to be the same. Since most lights in the multiple light scene are further away from the work surface than the single light source, the scene appears darker, though more evenly illuminated. The effect of the specular reflections is prominent on the lightbulb in the lower part of the image. In the single light source, there is a single specular reflection, while in the multiple light scene, the reflection of the array of lights is visible.

With the same camera setup for both scenes, we train NeRF models with the same number of iterations. We show the depth rendering in Fig. 7 and circle a glass and a wineglass on their side. In the single-light source image, the closer surfaces of the glasses are missing, while in the multiple-light source depth image, the glasses are nearly fully recovered. This suggests that additional lights in the scene can help NeRF recover the geometry of transparent objects better.

## 5.6 Workcell Setup

We experiment with a potential setup for a robot workcell in which a grid of overhead cameras captures views of the cluttered scene so that a robot manipulator arm can then perform tasks with



| 9 Cameras | 16 Cameras | 25 Cameras | 36 Cameras | 49 Cameras |

Figure 9: **Depth rendering using a grid of overhead cameras.** Using increasing amounts of overhead cameras improves the quality of the depth map and its utility in grasping, however, beyond a certain number of cameras there is a diminishing return.

transparent objects in the workcell. We propose that a grid of overhead cameras would be practical to setup and would not obstruct manipulator tasks nor operator interventions. The objective is to determine how many overhead cameras would be needed to recover a depth map of sufficient accuracy to perform manipulation tasks.

We place a 2 m by 2 m grid of cameras 1 m above the work surface, and have them all point at the center of the work surface. Each camera has the same intrinsics, and are evenly spaced along the grid. We experiment with grids having 4, 9, 16, 25, 36, and 49 cameras. The environment has the same 5x5 grid of lights as before. For each camera grid, we train JaxNeRF for 50k iterations and compare performance.

After training, we observe increasing peak signal to noise ratios (PSNR) and structural similarity (SSIM) scores with increasing number of cameras. The 2x2 grid of cameras produces a high train-to-test ratio for PSNR, likely indicating overfitting to training data, and results in a depth map without apparent geometry. This ratio decreases with additional cameras. The minimum number of cameras for this proposed setup appears to be around 9 (3x3) as its depth map is usable for grasp planning, while the 5x5 grid shows better PSNR and SSIM and ratio between train and test PSNR, and the 7x7 grid is the best. See Fig. 9 for a visual comparison. Additionally, we trained 9x9, 11x11, and 13x13 grids, observing no statistically significant improvement beyond the 7x7 grid.

## 6   Conclusion

In this work, we showed that NeRF can recover the geometry of transparent objects with sufficient accuracy to compute grasps for robot manipulation tasks. NeRF learns the density of all points in space, which corresponds to how much the view-dependent color of each point contributes to rays passing through it. With the key observation that specular reflections on transparent objects cause NeRF to learn a non-zero density, we have Dex-NeRF recover the geometry of transparent objects through a combination of additional lights to create specular reflections and thresholding to find transparent points that are visible from some view directions. With the geometry recovered, we pass it to a grasp planner, and show that the recovered geometry is sufficient to compute a grasp, and accurate enough to achieve 90 % and 100 % grasp success rates in physical experiments on an ABB YuMi robot. We created synthetic and real datasets for experiments in transparent geometry recovery, but we believe these datasets may be of interest to researchers interested in extending NeRF capabilities in other ways and thus contribute them as well. Finally, to test if NeRF could be used in a robot workcell, we experimented with grids of cameras facing a worksurface and their ability to recover geometry in potential setup, and showed the increased capabilities and point of diminishing return for additional cameras.

In future work, we hope to address one of the main drawbacks of NeRF—the long training time required to obtain a NeRF model. Many research groups have sped up training time through improved implementations, new algorithms, new network architectures, pre-conditioned network weights, focused sampling, and more. While these approaches apply to general NeRF training, we plan to exploit features specific to robot scenerios to speed up training, including using depth camera data as additional training data, manipulator-arm-mounted cameras to inspect regions of interest, and visio-spatial foresight to adapt to changes in the environment.

## Acknowledgments

This research was performed at the AUTOLAB at UC Berkeley in affiliation with the Berkeley AI Research (BAIR) Lab, Berkeley Deep Drive (BDD), the Real-Time Intelligent Secure Execution (RISE) Lab, and the CITRIS "People and Robots" (CPAR) Initiative. We thank our colleagues who provided helpful feedback and suggestions, in particular Matthew Tancik. This article solely reflects the opinions and conclusions of its authors and do not reflect the views of the sponsors or their associated entities.

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
