# OpenReview forum: "Dex-NeRF: Using a Neural Radiance Field to Grasp Transparent Objects"
_robot-learning.org/CoRL/2021/Conference — CoRL2021 Poster_

### Official Review · Reviewer_21Zr · 2021-07-19

**Originality:** Good
**Technical Quality:** Poor
**Clarity Of Presentation:** Good
**Impact:** 2

**Recommendation:**

Weak Accept: I recommend accepting the paper, but will not argue for my recommendation if the majority of other reviewers have a different opinion.

**Summary:**

This paper proposes to render depth maps of transparent objects with NeRF. The rendered depth maps can be used for grasping these transparent objects.

**Issues:**

As stated in weakness.

Also, the proposed method only uses reflection to locate the surface of the object. One important property of transparent objects, the distortion is not handled. If the method can utilize the clues of distortion, the few light sources would be enough to mode the geometry. That would be very interesting.

**Reviewer Expertise:**

Very good: Comprehensive knowledge of the area

**Strengths And Weaknesses:**

Strengths:
- NeRF can model transparent objects accurately. Based on the learned NeRF representation, depth maps can be rendered.
- This paper shows that using a threshold on density to locate boundary of the object and placing additional lights in the scene can improve the quality of the learned geometry.

Weakness:
- No quantitative comparisons. For example, the quantitative comparison between the depth maps rendered by vanilla NeRF and the proposed method (in simulation), and the grasp success rate given depth maps from RealSense, vanilla NeRF, and the proposed method (in simulation and real-world).
- There's not much novelty in the method part. It's just utilizing an existing method (NeRF) to a relatively new problem (render depth map of transparent objects).

**Summary Of Recommendation:**

It's an interesting idea to render depth maps of transparent objects with NeRF. However, the method is only slightly different from NeRF in depth rendering part. Most importantly, no quantitative experiments are conducted to prove the validity of the method. Only qualitative results are shown. It's hard to judge the quality of the obtained results.

Given the authors' concrete response, I would like to change to weak accept.

---

> ### Author Response · Authors · 2021-08-28
> **Thank you for taking the time to review the paper and provide valuable feedback, reviewer 21Zr.**
>
> Thank you for taking the time to review the paper and provide valuable feedback.
>
> # Novelty
> We were unsure and pleasantly surprised that NeRF-GTO was capable of generating geometric information of sufficient quality that it would be possible to compute reliable grasps on transparent objects.  As this is known to be a difficult task in robotics, we are excited about the implications of the result.  Additionally, we show in the updated results that using a vanilla-NeRF for grasping leads to very low success rates ($\le$ 30% with one exception), while the proposed modifications lead to $\ge$90% success rates for all tested objects.
>
> # Experimental Results
> We have thoroughly updated the results and include both physical experiments with an ABB YuMi, and simulated results using a physics-based simulation of grasp wrench resistance.  Please see the note to AC and/or the project website for more details (https://sites.google.com/view/nerf-gto/).
>
> In physical experiments, we compute 10 grasps each on different objects and compare NeRF-GTO to two baselines. NeRF-GTO is able to achieve a high success rate, 90% or better on all objects. In contrast, the first baseline of using a high-quality depth camera (PhoXi) is unable to recover any meaningful geometry of transparent objects from which to perform grasp analysis and results in 0% success rates.  The second baseline of using vanilla-NeRF is able to recover geometry, but that geometry often has unpredictable offsets leading to bad grasp predictions, and results in at most a 30% success rate, with a lone exception of an upright bottle.
>
> In simulation experiments, we plot grasp success rate vs number of training iterations and show that additional training leads to better success rates, plateauing at around 50k to 60k iterations.

---

> > ### Comment · Reviewer_21Zr · 2021-08-28
> > **Two more issues**
> >
> > The new results are really impressive. Just want to check the settings of grasp experiments:
> > 1. Do you train the dex-net grasp planner on your own dataset or do you just use the pre-trained weight?
> > 2. When comparing with NeRF, are you using the same network and weights but different depth rendering methods?

---

> > > ### Author Response · Authors · 2021-08-28
> > > **RE: Two more issues**
> > >
> > > Thanks! To answer your questions:
> > > 1. We are using pre-trained weights for Dex-Net.
> > > 2. Yes, we train NeRF once and use the same weights for both the vanilla NeRF baseline and NeRF-GTO. These results are to test the importance of the proposed depth rendering.

---

### Official Review · Reviewer_hn8F · 2021-07-22

**Originality:** Very Good
**Technical Quality:** Very Good
**Clarity Of Presentation:** Very Good
**Impact:** 4

**Recommendation:**

Weak Accept: I recommend accepting the paper, but will not argue for my recommendation if the majority of other reviewers have a different opinion.

**Summary:**

This paper tackles the problem of grasping transparent objects by using combination of NeRF and dexnet. It takes in multiple view of the scene and trains a nerf model that underneath predicts occupancy probability of each point in the workspace volume. They also add some small modifications to nerf where it reasons about view independent occupancy and also instead of aggregating the occupancy probabilities along the ray it thresholds the occupancies and terminates the depths at the first location that has a probability higher than a threshold. From the predicted depth, they use dexnet to generate grasps and pick up the objects.


**Issues:**

See weaknesses above.

**Reviewer Expertise:**

Excellent: Expert knowledge on the topic of the paper

**Strengths And Weaknesses:**

Strengths:
- This is a systems paper on how to integrate different modules properly to solve the task. I like the idea of using nerf for depth prediction and using dexnet
- The idea of using more lights to improve the depth quality also makes sense.

Weaknesses:
- What is the success rate on the whole system in grasping real transparent objects?
- How long does the training of nerf takes?
- What are the failure cases look for depth prediction and (depth prediction and grasp generation) look like?
- As far as I understand NeRF, \sigma is a probability between 0 and 1 for likelihood of a point being occupied or not. However, the thresholds are all more than 1. I would like to get clarification on why all these numbers are greater than 1.
- video of real robot experiments missing.


**Summary Of Recommendation:**

I like the general idea of the paper but I think the paper focuses too much on just depth prediction and I'd like to see how authors responds to the points I raised above.

---

> ### Author Response · Authors · 2021-08-28
> **Thank you for the thorough review and insightful questions, reviewer hn8F**
>
> Thank you for the thorough review and insightful questions.
>
> In the revised version of the paper we include extensive experiments in simulation and physical environments.  The simulation experiments use a physics-based grasp analysis to compute wrench resistance of proposed grasps. In the physical experiments, an ABB YuMi grasps and lifts an object based on the process proposed in the paper.  Please see our comment to the AC and/or updates on the project website (https://sites.google.com/view/nerf-gto/) for additional details.  With that as context, we answer the questions:
>
> # Grasp success rate on real transparent objects
> The success rate on physical objects, attempting 10 grasps each, is 90% or better. We include a table for this. We include baselines of a (1) high-quality PhoXi depth camera that proves unable to recover any meaningful geometry from which to perform grasp analysis and gets 0% success, and (2)  vanilla-NeRF which recovers geometry but includes unpredictable offsets that result in poor grasp predictions, and thus does not exceed 30% success, with a single exception.
>
> # How long does the training of NeRF takes?
> We have clarified in the revision that training time is quite long, usually minimum about 1 hour on a V100 GPU, though we often train for longer to test as close to ideal settings as possible. As the exact amount of training is an open issue, we also experiment in simulation to test grasp-success relative to training iteration--this too is included in the revised results.  As you might guess--longer training does improve grasp success, though plateaus around 50k or 60k iterations. Due to the long training time, we propose several promising directions to speed up NeRF training in future work.
>
> # Depth prediction failure cases
> The most common failure cases we see are (1) holes in the geometry, usually at the top of a cylindrical object on its side, and (2) filled geometry, e.g., an empty cup shows as full. The first issue appears to be the result of light placement and the resulting specular reflections--this we showed can be addressed with additional lights in the scene. We hypothesize that the second issue is caused by an insufficient number of cameras placed to see the inside of the cup.  We also observe that physical results often seem better than simulation--this is likely due to physical objects having imperfections providing a better signal from which NeRF can learn the geometry.
>
> # $\sigma$ between 0 and 1
> This is a great point to raise--we included the following discussion to help clarify.
>
> The value of $\sigma$ is not actually a probability or opacity/alpha value in the traditional sense, and the network architecture places no bounds on $\sigma$. The loss function converts $\sigma$ to a value between 0 and 1 by exponentiating it: $\alpha = 1 - e^{d\sigma}$, where $d$ is the distance between integration times along the ray.
>
> # video of real robot experiments missing.
> We will include the video.

---

### Official Review · Reviewer_nhV7 · 2021-07-24

**Originality:** Fair
**Technical Quality:** Fair
**Clarity Of Presentation:** Good
**Impact:** 2

**Recommendation:**

Weak Reject: I recommend rejecting the paper, but will not argue for my recommendation if the majority of other reviewers have a different opinion.

**Summary:**

The paper presented a method on recovering depth of transparent objects with "Neural Radiance Field", and the potential application in grasping. While the recovering depth part showed meaningful results, the paper does not provide much data point on the application of grasping, thus the main focus is on depth recovering.

**Issues:**

1 grasping should include more details, e.g. examples, experiments, metrics, so that reader could see the difference between having the proposed approach and baseline.
2 more details on experimental setup, e.g. how the cameras are configured, what metrics has been used. more quantitive results on a set of defined metrics.

**Reviewer Expertise:**

Good: General knowledge of the area

**Strengths And Weaknesses:**

Strengths: adopted NRF to recover depth of transparent objects, which is usually hard for conventional approaches.

Weaknesses: Does mention grasping in the title but not providing sufficient amount of data in experiments.

**Summary Of Recommendation:**

Unless provide more data on grasping or switch to focus purely on depth recovering, I would suggest on weak reject.

---

> ### Author Response · Authors · 2021-08-28
> **Thank you for the valuable feedback, reviewer nhV7**
>
> Thank you for the valuable feedback.
>
> # Depth Recovery and Grasping
> The reviewer’s point here is well taken.  We have updated the grasping results and include metrics on both simulated grasps and on physical grasps using an ABB YuMi robot.  In the simulated experiments, we use a physics-based grasp analysis software that computes wrench resistance for a given grasp on a given object.  In the updated results, in simulation we show the effect of training epoch on simulated grasp quality.
>
> ## Physical Experiments
> In physical experiments, we use Dex-Net to compute 10 different grasps each on 6 different transparent objects.  We then have the YuMi perform the grasp and lift the object.  We count successes when the object is lifted.  We have also included a baseline of using a high-quality overhead depth camera (PhoXi) and a vanilla-NeRF. For a detailed summary, please see our comment to the AC and/or the updates on the website: https://sites.google.com/view/nerf-gto/.  Updated regions on the website have a gray background.
>
> In physical experiments, NeRF-GTO was able to achieve a 90% or better grasp success rate on all objects, while the overhead depth camera and vanilla-NeRF were unable to exceed a 30% success rate with a single exception. The overhead camera normally provides very high quality depth images, but is unable to recover depth for transparent objects, giving no meaningful information on which to perform grasp analysis. The vanilla-NeRF was able to recover geometry, but had unpredictable offsets that lead to grasp predictions that did not succeed.
>
> ## Simulation Experiments
> In simulation experiments, we plot the results of a physics-based simulation of grasp wrench resistance for a given training epoch. The plot shows that training improves grasp success rate, plateauing after around 50k to 60k iterations.
>
> We also observe that physical results appear to be better than those in simulation.  We hypothesize that this is due to real transparent objects having imperfections that lead to more signals from which NeRF-GTO can learn the geometry.

---

### Official Review · Reviewer_qjfP · 2021-07-29

**Originality:** Good
**Technical Quality:** Good
**Clarity Of Presentation:** Excellent
**Impact:** 4

**Recommendation:**

Weak Accept: I recommend accepting the paper, but will not argue for my recommendation if the majority of other reviewers have a different opinion.

**Summary:**

This paper applies the wildly popular and promising NeRF -- wherein we learn a radiance field network that is integrated along a ray, resulting in estimates of the RGB and depth -- to the setting of transparent object detection for robotic grasping. For the latter, the paper makes use of another existing and widely-used method, DexNet.

The main contribution of the paper is showing that depth maps produced by NeRF integration are of sufficient high-quality for robotic grasping. This is important because typical 3D perception sensors relying on infrared imaging do a very poor job at handling the specularities and transparent nature of glass, the depth maps they output are almost unusable for manipulation. A secondary contribution is the creation of datasets of transparent objects, both synthetic and real.

**Issues:**

It would be great if weaknesses W2-W6 could be addressed by the rebuttal deadline. They'd place the paper in much better context.

**Reviewer Expertise:**

Very good: Comprehensive knowledge of the area

**Strengths And Weaknesses:**

Strengths

S1. The paper is well written and it addresses a very important and challenging problem in a practical way.

S2. It confirms that the NeRF depth map is practical for depth-based grasping and manipulation, which is very important.

S3. It contributes to addressing the issue of data collection for pose detection and grasping of transparent objects, without a need for object models and without a lot of need for manual handling of the objects in the scene. I think this is also very important.

Weaknesses

W1. Technically speaking, the proposed method is not novel. It "just" combines two existing methods. In fact, the original NeRF paper showcased a few examples of how it worked with transparent objects. That said, I think the whole is more than the sum of its parts here, and the paper offers many insights and experimental results about the performance of NeRF in a wide variety of transparent objects. I think this worthwhile enough for me to completely ignore the issue of novelty.

W2. I'd have appreciated some more details about the size of the 3 contributed datasets and how they differ from other papers proposing similar datasets.

W3. I'd have appreciated some discussion on the relationship between NeRF and ray-tracing methods. Can NeRF handle second-order light effects from ray bounces and shadows? What happens when you have transparent objects occluding other transparent objects?

W4. Section 4.2 was quite unclear to me. After reading it, it was still ambiguous whether the proposed method uses view-independent or view-dependent volume density sigma?

W5. It's not very clear how or whether the ground truth object pose was collected in the Cannon EOS datasets.

W6. The future work directions are mostly about speeding up the training NeRF, and not necessarily about increasing its accuracy. I'd have appreciated some discussion in that direction.

**Summary Of Recommendation:**

I am recommending that the paper be accepted. It addresses a very important and challenging problem in a way that alleviates the need for calibrated data collection, which is a very tedious process. While I would have liked some more extensive discussion on W3, W4, and W6, I find the paper to be useful enough to push this sub-area of robot perception forward.

---

> ### Comment · Reviewer_qjfP · 2021-08-28
> **one more issue**
>
> W7. It would be good to have a comparison with Cleargrasp [13], which is currently one of the main contenders in this space. The comparison could be in terms of depth map estimation or the quality of estimated surface normals.

---

> ### Author Response · Authors · 2021-08-28
> **Thank you for this insightful feedback, reviewer qjfP**
>
> # Novelty
> We were unsure and pleasantly surprised that NeRF-GTO is capable of generating geometric information of sufficient quality to compute reliable grasps on transparent objects.
>
> # Size and Difference of Datasets
> To clarify and update the list of datasets we provide the following table.  For a visualization, please refer to the project website: https://sites.google.com/view/nerf-gto/
>
> * Simulation (Blender)
>   * Clutter with light array
>   * Clutter with single light
>   * Singulated Pipe Connector
>   * Singulated Pawn
>   * Singulated Turbine Housing
>   * Singulated Mount 1
>   * Singulated Wineglass on side
>   * Singulated Wineglass, Upright
> * Physical
>   * Table with clutter
>   * Dishwasher
>   * Tape Dispenser
>   * Wineglass
>   * Flask
>   * Safety Glasses
>   * Bottle
>   * Lion Figurine in clutter
>
> We include 8 synthetic datasets of transparent objects generated with photorealistic settings in Blender. We note that these are computationally demanding to reproduce, due to the demands that refraction places on the renderer (some datasets required over a week to generate using a V100 GPU).
>
> We include 8 physical datasets of transparent objects with a variety of camera poses.  The main difficulty in generating these datasets is calibrating and computing high-precision camera poses.
>
> The datasets differ from other datasets in their focus on scenes with transparent objects in a graspable setting, with 70+ camera poses each.
>
> We will include these details in Section 4 of the revised submission.
>
> # NeRF and Second-Order Effects
> We observe that vanilla-NeRF does not support transparent object effects--it casts a single ray per source image pixel without reflection or bouncing. It handles shadows without issue. The incorporation of the view direction in its regression allows recovery of non-Lambertian effects that one might see on a rough but specular surface. However, when it comes to transparent objects, the usual result we observe is that the transparent object shows up as a “ghostly” or “blurry” version of the original object. We hypothesize that this is due to the neural network generalizing over the colors it sees when training. We clarified this point in Section 4.
>
> # View-dependent Sigma Clarification
> We use NeRF-GTO with a view-independent sigma in training, similar to vanilla-NeRF. We found that with the view-dependency, the quality of the recovered geometry suffered. As this is consistent with the ablation from the original NeRF we did not include the results of our informal ablation study confirming this.
>
> # Ground-truth poses in physical
> We used a combination of in-scene AR markers with known sizes and locations, and the bundle adjustment method from Colmap to recover the camera poses. We used physical measurements to recover ground-truth poses of objects relative to the AR markers. We clarified this point in Section 4.
>
> # Comparison with ClearGrasp
> We considered the ClearGrasp datasets, but they do not include the number of views required to train a NeRF model, so we were not able to benchmark using their data or directly compare to their results. We highlight a key difference between ClearGrasp and NeRF-GTO: ClearGrasp requires pre-training on objects in order to compute grasps and is not able to pick up out-of-distribution objects, while NeRF-GTO appears to work on a wide variety of objects without pre-training.
>
> # Future work
> Thanks for the great suggestion. We have found that the accuracy in our experiments, with sufficient camera coverage, has been high enough to generate high-quality grasps. There’s also an unusual disparity between the simulation and physical datasets--NeRF-GTO appears to do better with the physical datasets. We hypothesize that this is mainly due to physical objects having imperfections that aid in training the NeRF-GTO model. We think that an interesting direction for this would then be to determine additional factors that contribute to the quality of transparent object geometry recovery--especially as related to light sources. Additionally, we have included a result that simulates grasping at different training iterations (see update on website)--which is both a study of training speed and accuracy.

---

### Meta-Review · Area_Chair_DFrK · 2021-08-11

**Recommendation:** Accept (Poster)
**Confidence:** 4

**Metareview:**

The paper's contributions in regards to depth estimation have been well received, both in terms of its methodology and on the contribution of datasets. The writing is clear and the reporting of the investigations is broad and insightful. The methodology is sensible and original while its motivation is easy to understand.

However, it seems there is confusion when it comes to robotics evaluation. One interpretation is that there was no real grasping experiment at all, although Figure 1 implies that Dex-Net was used for the Fetch robot (if Fig. 1 is a simple illustrative figure this should be made clear). Assuming that no real-robotics experiments were done then sentences such as "Experiments on synthetic and real images showing NeRF with Dex-Net generates high-quality grasps." and "Grasping real-world Objects" are not supported with experimental, quantitative evidence (e.g. success rates). On the other hand, if real grasping experiments were done, then the paper is missing deeper evaluations which should be added or at least clarified during the rebuttal.

Post-rebuttal ====================

The authors clarified the main concerns and weaknesses during the meta-review. As a result, the reviewers have been positive in raising their scores. The authors now provide strong experimental evidence on the efficacy of their method.

---

> ### Author Response · Authors · 2021-08-28
> **Thank you to AC and all reviewers**
>
> We sincerely thank the reviewers and the AC for their time and diligence in reviewing our submission.  It is clear that each reviewer spent time to offer helpful suggestions, raise questions, and request clarifications, which has improved our revised submission.
>
> To address the main point of concern, we have expanded and clarified the experimental results, both in simulation and with physical grasps. Plots of these results are included in the updated (and still anonymized) website for the paper: https://sites.google.com/view/nerf-gto/. To speed up reviewing, we set the background of new/updated sections to gray. We summarize the experimental results here, and have included detailed comments for each reviewer.
>
> # Physical Experiments
> For physical experiments, we include 5 singulated transparent objects: Tape Dispenser, Wineglass, Flask, Safety Glasses, and Bottle; and 1 object in clutter: Lion Figurine. For each of these 6 objects, we used 1080p RGB images from a RealSense camera, recovering and refining camera poses as we did with the EOS dataset. After training the NeRF model, we compute 10 different grasps using Dex-Net, and attempt each grasp using an ABB YuMi robot. We consider a grasp successful if the YuMi is able to grasp and lift the object.  We compare to two baselines: (1) a high-quality overhead RGBD camera (PhoXi), and (2) vanilla-NeRF. The main difference between vanilla-NeRF and NeRF-GTO is method use to render the depth image that is fed to Dex-Net.
>
> ## Physical Grasp Success rates:
> | Object        | PhoXi |  Vanilla NeRF | NeRF-GTO |
> | :---           | ---: |  ---: |  ---: |
> | Tape Dispenser | 0/10 |  0/10 | 10/10 |
> | Wineglass      | 0/10 |  0/10 |  9/10 |
> | Flask          | 0/10 |  1/10 |  9/10 |
> | Safety Glasses | 0/10 |  0/10 | 10/10 |
> | Bottle         | 0/10 | 10/10 | 10/10 |
> | Lion Figurine          | 0/10 | 3/10 | 10/10 |
>
> We observe that the PhoXi overhead RGBD camera is unable to recover any meaningful depth information for the transparent objects. Without any depth signal on the object’s geometry, Dex-Net is unable to compute a grasp. Vanilla-NeRF is able to recover the geometry of the transparent objects, but the depth information around the transparent object is almost always offset unpredictably leading to low grasp success rates for all objects except the bottle.  In contrast, NeRF-GTO is able to recover geometry sufficient to achieve a high success rate in grasping all objects.
>
> # Simulation Experiments
>
> For simulation experiments, we follow a similar approach to the physical experiments, substituting a physics-based simulation of wrench resistance for each grasp. Here we are interested in how training time affects the ability to compute a grasp, and we simulated and plot grasp successes over training iterations. We observe that grasp-success rates increase with additional training, plateauing at around 50k to 60k training iterations. Please see the website for the plot.

---

### Decision · Program_Chairs · 2021-09-13

**Decision:**

Accept (Poster)

**Comment:**

The paper's contributions in regards to depth estimation have been well received, both in terms of its methodology and on the contribution of datasets. The writing is clear and the reporting of the investigations is broad and insightful. The methodology is sensible and original while its motivation is easy to understand.

However, it seems there is confusion when it comes to robotics evaluation. One interpretation is that there was no real grasping experiment at all, although Figure 1 implies that Dex-Net was used for the Fetch robot (if Fig. 1 is a simple illustrative figure this should be made clear). Assuming that no real-robotics experiments were done then sentences such as "Experiments on synthetic and real images showing NeRF with Dex-Net generates high-quality grasps." and "Grasping real-world Objects" are not supported with experimental, quantitative evidence (e.g. success rates). On the other hand, if real grasping experiments were done, then the paper is missing deeper evaluations which should be added or at least clarified during the rebuttal.

Post-rebuttal ====================

The authors clarified the main concerns and weaknesses during the meta-review. As a result, the reviewers have been positive in raising their scores. The authors now provide strong experimental evidence on the efficacy of their method.